# Diversify Your Vision Datasets with Automatic Diffusion-Based Augmentation

**Lisa Dunlap**   **Alyssa Umino**   **Han Zhang**   **Jiezhi Yang**   **Joseph E. Gonzalez**   **Trevor Darrell**

UC Berkeley

## Abstract

Many fine-grained classification tasks, like rare animal identification, have limited training data and consequently classifiers trained on these datasets often fail to generalize to variations in the domain like changes in weather or location. As such, we explore how natural language descriptions of the domains seen in training data can be used with large vision models trained on diverse pretraining datasets to generate useful variations of the training data. We introduce ALIA (Automated Language-guided Image Augmentation), a method which utilizes large vision and language models to automatically generate natural language descriptions of a dataset's domains and augment the training data via language-guided image editing. To maintain data integrity, a model trained on the original dataset filters out minimal image edits and those which corrupt class-relevant information. The resulting dataset is visually consistent with the original training data and offers significantly enhanced diversity. We show that ALIA is able to surpasses traditional data augmentation and text-to-image generated data on fine-grained classification tasks, including cases of domain generalization and contextual bias. Code is available at `https://github.com/lisadunlap/ALIA`.

## 1 Introduction

While modern pretraining data are incredibly diverse, datasets for specialized tasks such as fine-grained animal identification are often much less so, resulting in trained classifiers that fail when encountering new domains such as a change in weather or location. An effective method to address this is to add more training data from the test domain (37), but obtaining and labeling this additional data is often costly and it can be challenging to determine which domains to gather data from.

To address this, recent works have utilized image generation models trained on large pretraining datasets to supplement incomplete training sets by generating diverse examples from generative models fine-tuned on the training set (8; 14; 1). Furthermore, previous work in language-guided data augmentation with vision and language models relies on user-supplied domain descriptions (7) or descriptions generated from word-to-sentence models (9).

Although these methods can be effective in increasing image diversity, they either require finetuning the image generation model, which can be prohibitively expensive, or generating images which have no visual grounding in the original training data. While the latter may be suitable for common benchmarks, such as ImageNet (6), it proves to be much less effective when we move to a specialized setting where generative models like Stable Diffusion (30) cannot recreate images which resemble the training data from text alone, as shown in Figure 1. In our work, we focus on how to utilize pretrained vision and language models for image captioning and generation as a *translation* layer between task-specific image data and task-agnostic natural language descriptions of domains. Since these high-level domain descriptions are well-represented by image generation models like Stable Diffusion, we can use them to perform *language-guided image editing* of the specialized training data.

37th Conference on Neural Information Processing Systems (NeurIPS 2023).

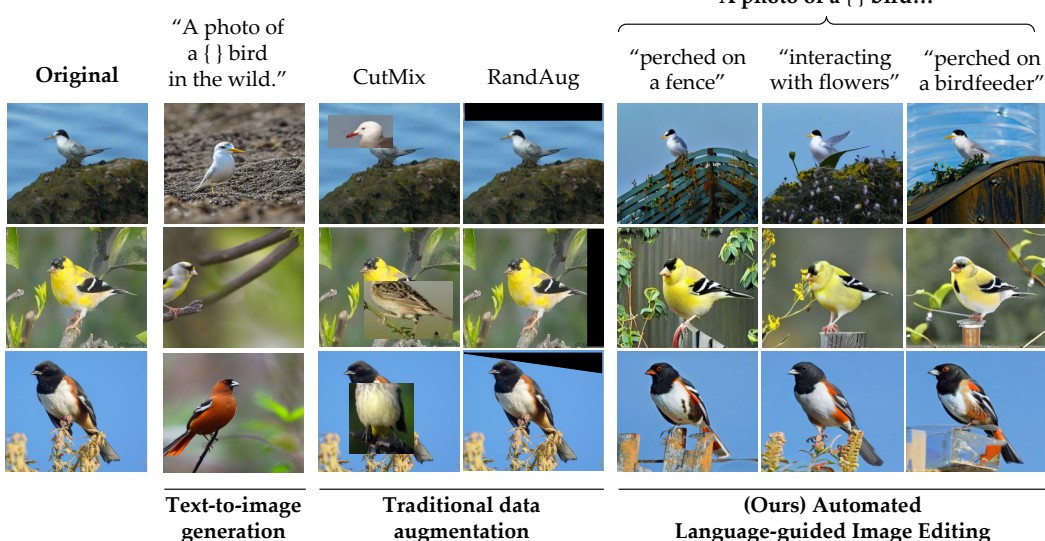

Figure 1: **Overview.** Example augmentations using text-to-images generation, traditional data augmentation methods, and our method, Automated Language-guided Image Editing (ALIA) on CUB (41). Images generated by ALIA retain task-relevant information while providing more domain diversity as specified by the prompts. Within the prompts, { } indicates the specific class name.

This produces images which are visually consistent with the training data, vary the task-agnostic domains, and preserve the task-relevant information present in the original image.

Specifically, our method ALIA (Automated Language-guided Image Augmentation) first generates captions for each image, summarizes the captions into a short list of domain descriptions with a large language model (LLM), and then uses these descriptions to generate edits of the training data with Stable Diffusion. To ensure data quality, we use a classifier trained on our original training set to remove images which (1) do not change the domain of the original image as desired or (2) corrupt task-relevant information. After filtration, we are left with an edited dataset visually consistent with the original data and representing all domains seen in testing (Figure 1). ALIA does not require finetuning the image captioning or image generation model, nor does it require user-supplied prompts.

We evaluate on fine-grained bird classification (CUB (41)), domain generalization (iWildCam (17)), and contextual bias (Waterbirds (32)) datasets. We show that the addition of ALIA generated data outperforms traditional data augmentation techniques and text-to-image generated data by up to 7%, even beating the performance of adding in real data on iWildCam. Furthermore, we investigate how our domain descriptions produce more useful edits than user-provided prompts, and examine the effect of filtering and the choice of image editing techniques (Section 5.4).

## 2 Related Works

**Supplementing Training Data with Generative Models.** Using generative models for data augmentation is a well-explored area of research, specifically in medicine (8; 33), domain adaptation (12) and bias mitigation (34). These methods train or use a pretrained GAN to generate images from the desired distribution. Furthermore, GANs have been used to supervise dense visual alignment (28) and generate pixel-level annotations from a small amount of labels (45; 18; 39). Recently, several works (9; 35; 38) have shown diffusion models' ability to generate training data in zero or few shot settings as well as generate hard training examples (13). While these works do show the promise of diffusion-generated data, models trained on diffusion-generated data obtain significantly worse accuracy than models trained on real datasets unless finetuned for that specific task (1; 38). In contrast, we use diffusion models to do *image editing* with text rather than generating images from text alone, resulting in augmentations that closely resemble the training data without finetuning.

**Traditional Data Augmentation.** Traditionally, data augmentation techniques involve random flipping, cropping, color shifting, etc. to manipulate the original images and create new versions of these images (36). More recently proposed mixup-based data augmentations aim to make the augmented data more diverse by cutting and pasting patches of two input images (42), applying convex combinations (43), and adaptively learning a sample mixing policy (10; 20; 15; 5). These methods often result in images that look unnatural as shown in Figure 1, while our method aims to create augmentations which are visually consistent with the original training data.

**Language-guided Data Augmentation for Classification.** Recent works (7; 9; 29) have explored how natural language descriptions of the domains and classes seen during testing can be used to improve performance and generalization in a few-shot or domain adaptation setting. These methods typically employ text-to-image generation with prompts sourced from language models (9), or augment training images using user-provided descriptions of both training and unseen test domains in a shared vision and language space (7). Similar to our work, these techniques use the high-level domain knowledge of large models to diversify image data with language, but crucially they either rely on user-provided domain descriptions or generate descriptions and augmentations that are not grounded in any real image data. In contrast, the foundation of our work is to ground both the domain descriptions and the generated training data in the given training set.

## 3 ALIA: Automated Language-guided Image Editing

Given a labeled training dataset, we aim to augment the dataset with images that are edited to improve the representation of various *domains*. We define *domain* to be any aspect of an image that is not intended to be used for classification (e.g. location, weather, time of day). The key insight of our method is to utilize image captioning and image generation models trained on large amounts of pretraining data to summarize the domains in the training set and use those descriptions to augment training data using text-conditioned image editing.

Our method consists of 3 stages: generating domain descriptions, generating useful image edits, and filtering out poor edits. An overview of our method is shown in Figure 2.

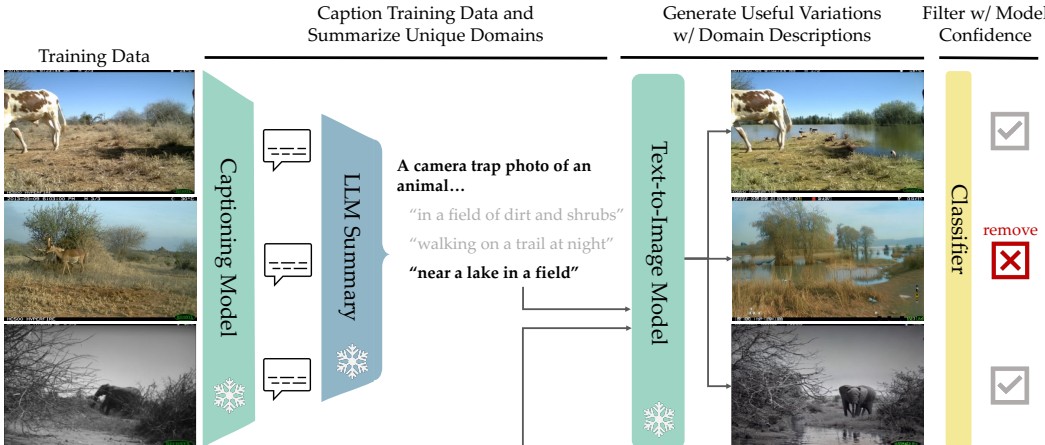

Figure 2: **ALIA.** Given a specialized dataset, we caption all images in our dataset using a pretrained captioning model, and feed these captions into a large language model to summarize them into a small (<10) set of natural language descriptions. Utilizing these descriptions, we perform text-guided image augmentation via a pretrained text-to-image model, thereby generating training images that align with the described settings. Finally, we apply two filtering techniques: a CLIP-based semantic filter to eliminate obvious edit failures, and a confidence-based filter that removes more subtle failures by filtering edits confidently predicted by a classifier trained on the original dataset.

### 3.1 Generating Domain Descriptions

A key idea of our method is to use captioning and language models trained on large amounts of pretraining data to summarize the potential domains. We assume knowledge of the superclass that

encapsulates all classes, such as "bird" for CUB or "animal" for iWildCam and a prefix to guide the format of the desired caption form, for example "a photo of a bird...". Once these prompts are generated, we found that we often achieve higher quality edits by adding the class name into the prompts after the fact (e.g. "a photo of a Scott Oriole bird...").

To generate a concise list of domain descriptions, we first caption each image in the dataset using a pretrained captioning model. This produces a comprehensive set of captions, which may highlight potential domains seen at test time. Note that these captions do not need to accurately describe the task-specific information, such as the species of the bird, as their purpose is to provide a broad overview of the context, such as the environment the bird is in or the actions of the bird. Additional image data of possible domains seen in testing but that don't contain any task-relevant information can also be used in this step. For example, when performing animal classification, one can add in background images of different locations that may be seen in deployment.

We assume that the amount of training data is not small ($<$ 100 samples). Therefore, we use an LLM to summarize these captions into a list of domains which are agnostic of the class. Due to constraints on context length, we randomly sample 200 unique captions and use the following prompt:

> "I have a set of image captions that I want to summarize into objective descriptions that describe the scenes, actions, camera pose, zoom, and other image qualities present. My captions are [CAPTIONS]. I want the output to be a handful of captions that describe a unique setting, of the form [PREFIX]"

We then ask a refinement question to ensure each caption is of only one setting and agnostic of the class with the following prompt:

> "Can you modify your response so each caption is agnostic of the type of [SUPER-CLASS]. Please output less than 10 captions which cover the largest breadth of concepts."

The end result is a list of less than 10 domain descriptions that often include domains seen in the test data (e.g. "A camera trap photo of an animal near a lake."). These descriptions serve as the foundation for the subsequent text-conditioned image editing stage of our method. We use BLIP (19) captioning model and GPT-4 (25) for summarizing the captions. A complete list of prompts for each dataset is given in Section 5.

## 3.2 Editing Images with Language Guidance

Once the domain descriptions are generated, they are used to condition the edits of original images in the training set. In our experiments, we employ two editing techniques based on Stable Diffusion; however, ALIA can be used with any text-conditioned image editing method. The two techniques deployed for our experiments are as follows:

*Image to Image with Text Guidance (Img2Img) (30; 2; 22)*: This technique first uses an image encoder to translate a provided image into a latent representation. Then, leveraging a diffusion model, this latent representation is progressively modified through a series of transformations, conditioned on the user-provided text. Finally, the modified latent representation is decoded to generate an augmented image that incorporates the desired modifications specified in the prompt.

*Instruct Pix2Pix (3)*: Given an edit instruction (e.g. "put the animals in the forest") and an image, this method generates a modified image adhering to the specified instruction. This is accomplished by training a conditional diffusion model on a dataset composed of paired images and edit instructions.

Among the many existing image editing techniques (23; 11; 4; 44; 26), we selected the two above for their ease of use and quality of outputs. Notably, the methods highlighted here share a common goal: to generate coherent, visually-grounded augmentations conditioned on the natural language descriptions extracted from the dataset.

## 3.3 Filtering Failed Edits Using Semantic and Visual Features

As depicted in Figure 3, there are three distinct failure cases for the text-conditioned augmentation: (1) total failure, where the edited image is vastly different from the original, (2) identity failure, where

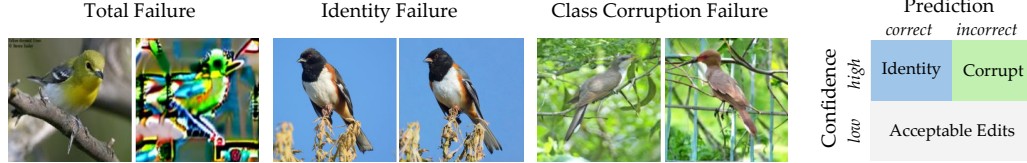

Figure 3: **Filtering.** Semantic filtering eliminates instances of total failure (left), but often misses instances of minimal edits (center) or edits which corrupt class information (right). Meanwhile, our confidence-based filtering mechanism is able to remove these failures by leveraging the prediction confidence of classifier trained on the original dataset applied to the image edits.

the edited image is nearly identical to the original, and (3) class corruption failure, where the edit significantly changed task-specific features. While previous work (9) utilized CLIP-based filtering to remove low-quality images, it only removes instances of total failure in settings where CLIP does not have a good representation of the classes. As such, we also employ a confidence-based filtering technique which uses a classifier trained on the original training set to determine instances of identity and class corruption failures.

*Semantic Filtering.* We use CLIP to predict whether the generated image is related to the task or not. For example, in the CUB dataset, we provide the text prompt "a photo of a bird" as well as the filtering prompts "a photo of an object", "a photo of a scene", "a photo of geometric shapes", "a photo", and "an image". All images that are not classified as "a photo of a bird" are removed.

*Confidence-based Filtering.* We take inspiration from previous work in identifying mislabeled examples using model confidence (24). After training a model $f$ on the original dataset, we calculate a confidence threshold $t_y$ for each class $y$ by averaging the softmax score of the correct label for each image in the training set. Specifically, given edited image $x'$ with label $y$ and prediction $\hat{y}$, it is filtered out if confidence$(f(x'), \hat{y}) \geq t_{\hat{y}}$. In the case that the image was correctly predicted ($y = \hat{y}$), since the model is already confident in its prediction, the edited image is likely to contain roughly the same information as the original, and thus should be filtered out. In cases where $y \neq \hat{y}$, high confidence suggests that the edit has corrupted the class to a point that it more highly resembles another class.

In our experiments, we randomly sample a portion of the filtered augmentations to incorporate back into the training set, leading to a dataset size expansion between 20-100%. Given that the prompts and the image edits are grounded in the original training set, ALIA is able to preserve visual consistency and encompass a broader array of domains.

## 4  Experimental setup

### 4.1  Implementation.

We fine-tune a ResNet50 for the CUB (41), iWildCam (17), and Waterbirds (32) datasets. We use the PyTorch pretrained models (27) on ImageNet with an Adam optimizer (16) and cosine learning rate scheduler. For each method, we do a hyperparameter sweep across learning rate and weight decay and choose the parameters with the highest validation performance. We train on 10 GeForce RTX 2080 Ti GPUs.

For all the diffusion-based editing methods, we use Stable Diffusion version 1.5 (31) from Hugging-Face (40) with the default hyperparameters aside from the edit strength (how much to deviate from the original image) and the text guidance (how closely the generated image should align with the text prompt). For these parameters, we search over 5 different values each for edit strength and text guidance, visualizing the resulting generations for a random sample (10 images) across 4 random seeds (40 generated images in total). We pick the parameters which generate the most diverse images that both retain the task-relevant information and remain faithful to the edit instruction. More details on this selection process including visualizations as well as results on training on data generated with different edit strengths and text guidance are in the Appendix. Results are averaged over 3 random seeds and further details on the hyperparameter search space and final choice of hyperparameters are listed in the Appendix.

## 4.2 Baselines.

The *Baseline* model is trained on the original training dataset, without any data augmentation. We additionally compare adding in the generated data with our method to the original training set (*+ALIA*), adding in real data from the test distribution (*+Real*), adding in diffusion generated data from text alone (*+Txt2Img*), and two traditional augmentation baselines (*+CutMix, +RandAug*). In order to perform a fair comparison, we keep the number of images added per class to the training set consistent across methods.

For the *+ALIA* and *+Txt2Img* results, we generate twice as much data as the original training set for each language prompt to ensure enough data is available after filtering. For *+ALIA*, this entails generating 2 edits for every image. We then randomly sample from these datasets to match the class distribution of the held-out real data used for the *+Real* baseline. For the *+Txt2Img* baseline, we generate images with a hand-crafted prompt for each dataset, and apply our semantic filtering to the generated images to remove low-quality samples. We provide the amount of data added and the prompts used for the *+Txt2Img* baseline in the subsequent sections.

Lastly, we compare to two data augmentation baselines taken from recent literature: (1) *CutMix* (42), which generates mixed samples by randomly cutting and pasting patches between training images to encourage the model to learn more localized and discriminative features and (2) *RandAugment* (5), an automated data augmentation technique that reduces the search space of the type and magnitude of augmentations to find the best image transformations for a given task. For these baselines, we use the implementations in the PyTorch Torchvision library (21).

# 5 Experiments

We evaluate on 3 specialized tasks: domain generalization via camera trap animal classification (iWildCam), fine-grained bird classification (CUB), and bird classification with contextual bias (Waterbirds). Details of each dataset are listed in subsequent sections, with more statistics in the Appendix.

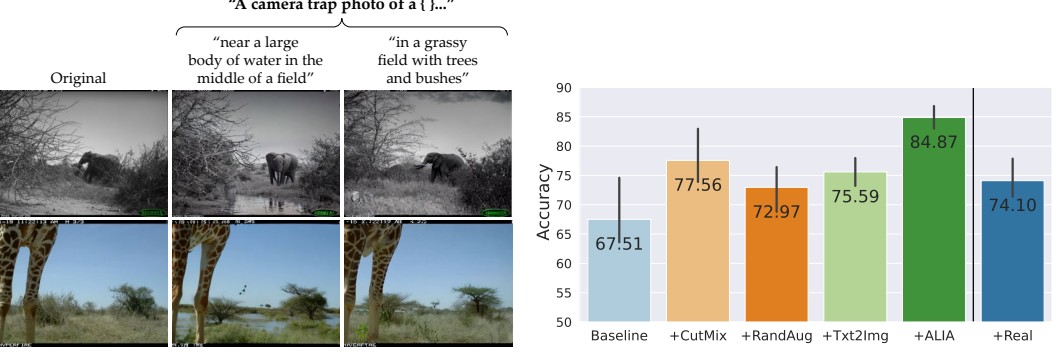

Figure 4: **iWildCam Subset.** Left plot visualizes the original training images and the corresponding data generated from ALIA based on the generated prompts. Right plot shows accuracy for adding in generated data from ALIA and baselines as in Section 4.2. ALIA significantly outperforms all the baselines including adding in real data from the test distribution, while +Txt2Img and +RandAug see smaller improvements.

## 5.1 Domain Generalization [iWildCam (17)]

The iWildCam dataset is a large-scale collection of images captured from camera traps placed in various locations around the world. We subsample the dataset to create a 7-way classification task (background, cattle, elephant, impala, zebra, giraffe, dik-dik), with 2 test locations that are not in the training or validation set. The training set has 6,000 images with some classes having as few as 50 images per example, and our *+Real* data baseline contains roughly 2,000 images from locations not seen during training. We generate our domain descriptions from the background images from the test domain, but do not use these images in training (details in Section 8.3). Since many images in the

original training set contain watermarks or timestamps while the text-to-image generated data usually does not, we crop the top and bottom of each image in prepossessing.

The prompts generated by ALIA are *"a camera trap photo of a { }..."*:
(1) *"in a grassy field with trees and bushes."*, (2) *"in a forest in the dark."*, (3) *"near a large body of water in the middle of a field."*, (4) *"walking on a dirt trail with twigs and branches."*.

We use *"a camera trap photo of a { } in the wild."* as the prompt for the *+Txt2Img* baseline.

As shown in Figure 4, ALIA not only outperforms all baselines, with a 17% performance improvement over training on the original data, but even exceeds the performance of adding in the same amount of real data. Note the improved performance over real can be somewhat attributed to that fact that our prompts are taken from empty images from the test domain rather than the training set.

## 5.2 Fine-grained Classification [CUB (41)]

CUB is a fine-grained bird classification dataset comprised of photos taken from Flickr. We remove 5 of the 30 images from each of the 200 classes from the training set for the *+Real* comparison.

The prompts generated by ALIA are *"a photo of a { } bird..."*:
(1) *"interacting with flowers."*, (2) *"standing by the waters edge."*, (3) *"perched on a fence."*, (4) *"standing on a rock."*, (5) *"perched on a branch."*, (6) *"flying near a tree, sky as the backdrop."*, (7) *"perched on a birdfeeder."*

We use *"an iNaturalist photo of a { } bird in nature."* as the prompt for the *+Txt2Img* baseline.

As shown in Figure 5, ALIA outperforms all the baselines aside from RandAug and adding in real data, while adding text-to-image generated data results in similar performance with the baseline. Significantly, these results show that ALIA provides performance improvements even in conditions devoid of domain shifts, thereby broadening its utility.

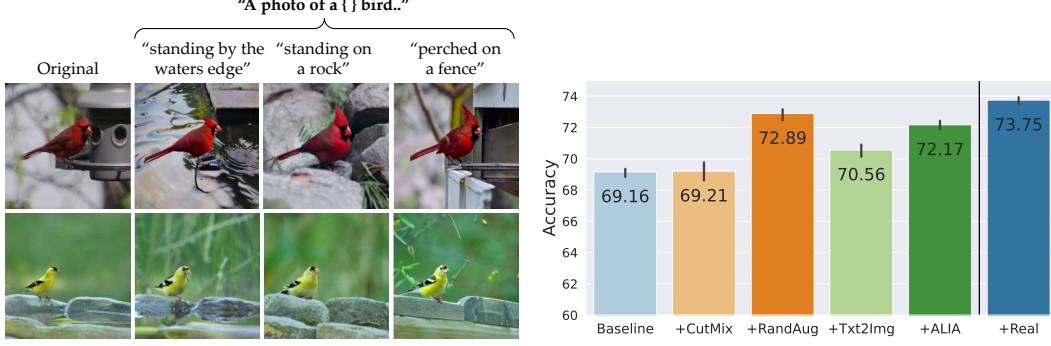

Figure 5: **Bird Classification (CUB).** Left plot visualizes the original training images and the corresponding data generated from ALIA based on the generated prompts. Right plot shows accuracy for ALIA and baselines. ALIA outperforms all the baselines aside from RandAug and adding in real data, while adding text-to-image generated data results in similar performance with the baseline.

## 5.3 Contextual Bias [Waterbirds (32)]

Waterbirds is a synthetically constructed dataset which creates contextual bias by taking species of landbirds and waterbirds from the CUB-200 (41) dataset and pasting them onto forest and water backgrounds from the Places (46) dataset. The classification task is landbirds versus waterbirds, and the spurious correlation is the background. The training and validation sets have all landbirds appearing on forest backgrounds and all waterbirds appearing on water backgrounds, while the test set has an even representation of backgrounds and bird types. Our *+Real* data baseline contains only images from the out-of-domain groups, namely landbird on water and waterbird on land.

Interestingly, we found that the edits generated by the Img2Img editing method were very low quality (visualizations in the Appendix). Thus, we use InstructPix2Pix, prompting the LLM with the prefixes "a photo of a bird" and editing the final prompts to fit the instructional format of the editing method.

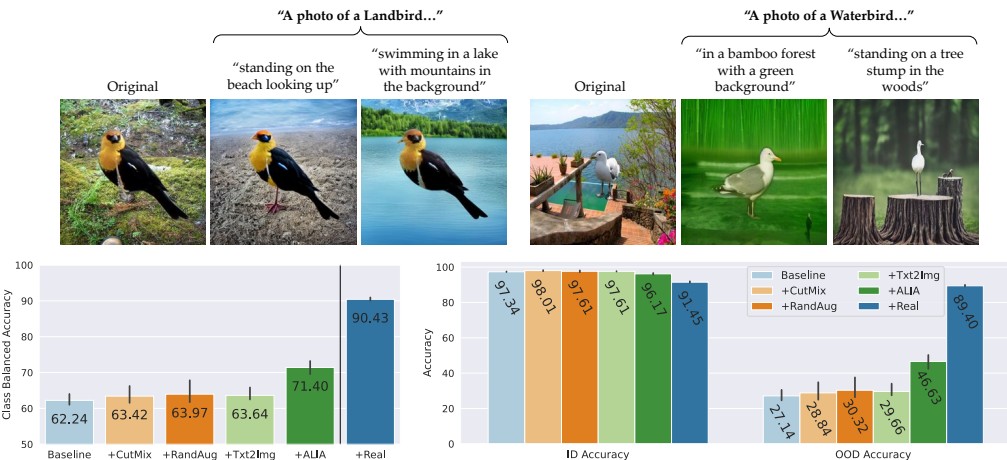

Figure 6: **Waterbirds Results.** Left plot shows class-balanced accuracy for ALIA and baselines. Right plot shows the breakdown of in-domain (waterbirds on water, landbirds on land) and out-of-domain performance (waterbirds on land, landbirds on water). ALIA is able to roughly match in-domain accuracy of the other methods while drastically improving out-of-domain accuracy of all other augmentation methods. This results in 7% improvement for class-balanced accuracy when compared to other augmentation baselines.

The prompts generated by ALIA are *"a photo of a { }..."*:
(1) *"in a bamboo forest with a green background."*, (2) *"flying over the water with a city skyline in the background."*, (3) *"perched on a car window."*, (4) *"standing in the snow in a forest."*, (5) *"standing on a tree stump in the woods."*, (6) *"swimming in a lake with mountains in the background."*, (7) *"standing on the beach looking up."*

We use *"an iNaturalist photo of a { } in nature."* as the prompt for the *+Txt2Img* baseline.

As shown in Figure 6, ALIA is able to approximately match the in-domain accuracy of the other augmentation methods while outperforming all other methods besides the real data baseline for overall accuracy. While ALIA's performance on this dataset is much lower than the state of the art methods, we believe that these results point to a promising future direction in how to modify ALIA to identify and amplify specific class-domain pairings which are underrepresented in the training set.

## 5.4 Ablations

**Effect of Prompt Quality and Filtering.** We ablate the quality of ALIA generated prompts by comparing the edit prompts generated by ALIA against those provided by users. For the user-supplied prompts, we draw on the engineered prompts that were used for the *+Txt2Img* baseline. As shown in Table 1, descriptions generated by ALIA outperform user-provided prompts, especially in the contextual bias setting, indicating that our generated prompts are able to accurately describe key domain-specific features and variations, resulting in more effective data augmentation for mitigating biases and improving model generalization. Moreover, Table 1 also demonstrates how our semantic and confidence-based filtering further improves accuracy.

| Dataset | User Prompt | ALIA Prompts | ALIA Prompts + Filtering |
|---|---|---|---|
| iWildCam | 79.92±4.22% | 82.57±2.19% | 84.87±1.92% |
| CUB | 71.02±0.47% | 71.25±0.86% | 72.70±0.10% |
| Waterbirds | 63.64 ±1.43% | 70.22±0.53% | 71.40±1.85% |

Table 1: **Effect of ALIA Generated Prompts and Filtering.**

**Choice of Image Editing Method.** As mentioned in Section 5.3, the suitability of image editing methods can vary based on the specific dataset. As such, we explore the impact of the editing method on performance by modifying the iWildCam subset from Section 5.1 with InstructPix2Pix. We employ the same prompts as in the Img2Img case, reformatted to the instructional syntax required by InstructPix2Pix (e.g. "a camera trap photo of a {} in a grassy field with trees and bushes" becomes "put the {} in a grassy field with trees and bushes"). Figure 7 presents examples of edits generated using InstructPix2Pix. Although InstructPix2Pix can produce edits with more significant color variations than Img2Img, it frequently applies an unrealistic airbrush-like effect or erases most of the information present in the original image. This outcome can be attributed to the fact that InstructPix2Pix is fine-tuned on a dataset of paired images and edits, which primarily consist of stylistic changes to artistic images, making the iWildCam images fall outside their domain. When comparing the performance of InstructPix2Pix, we find that it achieves an accuracy of $76.23 \pm 4.78\%$, compared to $84.87 \pm 1.92\%$ obtained using Img2Img edits. These results indicate that the choice of image editing method has significant impact on the final performance of ALIA, and thus it is crucial to profile the quality of image edits from various techniques on one's specific dataset.

**"put the {} near a large body of water in the middle of a field."**

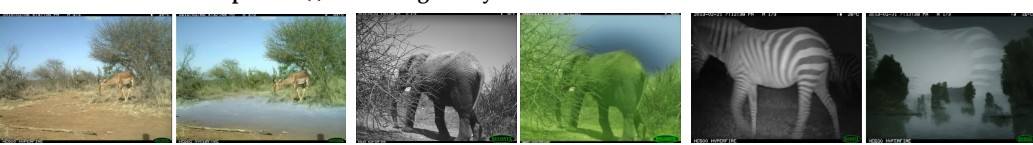

Figure 7: **InstructPix2Pix edits on iWildCam.** Despite offering more color variation, the method often imparts an artificial airbrush-like effect or removes substantial original image details.

**Amount of Augmented Data Added.** Figure 8 depicts the accuracy on CUB as a function of the number of generated images added where the grey line is the baseline accuracy. ALIA is able to achieve accuracy gains up to 1000 images (20% of the original training set), at which point accuracy starts to decline. In contrast, images generated from text alone see a decrease in accuracy almost immediately, reaching below the accuracy of the original training set at 2000 images. We suspect this is because much of the text to image data is from a different distribution than the regular training data, and thus adding small amounts can increase robustness while large amounts cause the model to overfit to this distribution. Since image to image edits use language to edit the training images directly, these images are less likely to be out of distribution as compared to the text to image data.

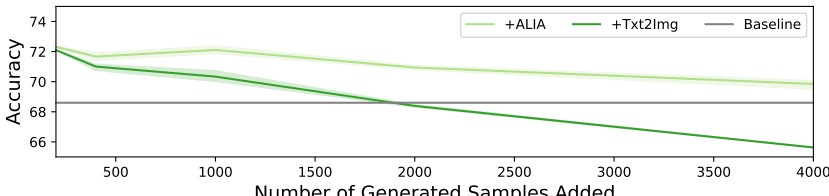

Figure 8: **Size of Augmented Dataset (CUB).** Results are shown for adding 200, 400, 1000, 2000, and 4000 generated images to the training set. The grey line depicts the baseline accuracy on the original 4994 images. ALIA is able to achieve accuracy gains up to 1000 images, at which point accuracy starts to decline. In contrast, images generated from text alone see a decrease in accuracy almost immediately, reaching below the accuracy of the original training set.

## 6 Limitations

While ALIA is able to generate impressive augmentations, there are limitations. As ALIA depends on large pretrained vision and language models to translate between task-specific image data and task-agnostic natural language descriptions of domains, performance of the method is bottlenecked by the quality of the captioning model, LLM, and image editing method (see Section 5.4). Furthermore, as we assume that the task-specific information in the training data cannot be easily generated via text alone, attempts to change aspects like the pose of a bird in CUB is likely to result in a failed edit. Finally, determining the optimal quantity of augmented data to reincorporate into the training set remains an unresolved question, and is an area we look forward to addressing in future research.

# 7 Conclusion

We present ALIA, a novel approach to data augmentation that leverages the high-level domain knowledge of large language models and text-conditioned image editing methods. By grounding both the domain descriptions and the augmented training data in the provided training set, our method has demonstrated impressive capabilities in several challenging settings, including domain adaptation, bias mitigation, and even scenarios without domain shift. Our findings offer exciting potential for future research, suggesting that the utilization of structured prompts generated from actual training data can be an effective strategy for improving dataset diversity. As the capabilities of captioning, LLMs, and image editing methods grow, we expect the efficacy and scope of our approach to increase.

**Acknowledgements.** We thank Suzie Petryk for her invaluable feedback on the manuscript. This work was supported in part by the NSF CISE Expeditions Award (CCF-1730628), DARPA's SemaFor, PTG and/or LwLL programs, and BAIR's industrial alliance programs.

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
