# Supplementary Material

## 8 Dataset Details

### 8.1 Image Counts

We provide the number of images for each dataset split used in Section 5 in Table 2, where Extra denotes the amount of data added for the real data baseline. This extra set was constructed for each dataset manually.

| Dataset | Training | Extra | Validation | Testing | % added |
|---|---|---|---|---|---|
| iWildCam | 6052 | 2224 | 2826 | 8483 | 37% |
| CUB | 4994 | 1000 | 5794 | 5794 | 20% |
| Waterbirds | 1139 | 839 | 600 | 5794 | 74% |

Table 2: Dataset Split Sizes. Note that CUB's validation set and test set are the same.

| | Landbird | | Waterbird | |
|---|---|---|---|---|
| | land | water | land | water |
| Train | 874 | 0 | 0 | 265 |
| Val | 467 | 0 | 0 | 133 |
| Extra | 0 | 650 | 189 | 0 |
| Test | 2255 | 2255 | 642 | 642 |

Table 3: Dataset Statistics for Waterbirds

### 8.2 Time to Generate Image Edits

Table 4 depicts the time to generate 2 image edits for each dataset, which are used in ALIA. The choice to generate twice as much data as the dataset was to ensure enough data was present after filtering to match the amount of images in the real data baseline. In practice one could edit a small percentage of their dataset and see what percentage of images were filtered out, then use that to determine roughly how many images to edit in order for the filtered dataset to be of the desired size.

| Generation Time (hours) | iWildCam | CUB | Waterbirds |
|---|---|---|---|
| | 7 | 6 | 2 |

Table 4: Dataset Generation Times

### 8.3 iWildCam Subset Construction

We chose to construct a subset of iWildCam because (1) the experimentation and generation time for the entire wilds dataset was prohibitively expensive and (2) some classes didn't have enough examples for us to split into a train, val, test, and extra set. While we did not artificially balance the classes, we did want classes which would have at least 40 examples in each split.

In an effort to keep this constrained setting as close to the original iWildCam dataset as possible, we constructed the train/val/test/extra splits such that each split contains non-overlapping locations and we did not subsample within locations to balance the class support.

Since iWildCam has a train, id_test, and test split already, we sampled the locations with at least two classes to put into our subset, ensuring that the train subset was sampled from the train set, the val subset was sampled from the id_test set. We split the sampled locations from the iWildCam test set into two disjoint groups, which formed the test set and extra set for our subset. These new splits were set before experimentation, so all methods were trained, validated, and evaluated on the same data. Figure 9 displays background images from the locations present in the test set and Table 5 displays the class counts per split.

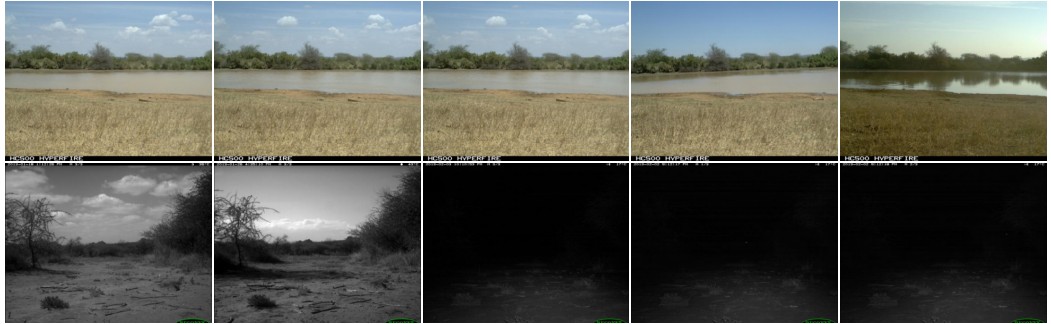

Figure 9: **iWildCam Test Locations.** Random samples from the two test locations that make up the iWildCam Subset test set.

| Split | Background | Cattle | Elephant | Impala | Zebra | Giraffe | Dik-Dik |
|-------|-----------|--------|----------|--------|-------|---------|---------|
| Train | 2210 | 801 | 366 | 981 | 720 | 460 | 514 |
| Val | 2006 | 206 | 53 | 140 | 119 | 58 | 244 |
| Test | 127 | 1416 | 2003 | 4553 | 144 | 47 | 213 |
| Extra | 402 | 506 | 91 | 625 | 85 | 47 | 468 |

Table 5: Class Counts for iWildCam Subset

# 9    Hyperparameters

To select the hyperparameters, we train the baseline with learning rates of [0.00001, 0.0001, 0.001, 0.01, 0.1] and weight decay [1e-5, 1e-4, 1e-3, 1e-2], selecting the configuration that results in the highest validation accuracy. These parameters, shown in Table 6, are then used across all methods.

| Dataset | Training | | | Img2Img | | InstructPix2Pix | |
|---------|----------|----|----|----------|----|-----------------|----|
| | learning rate | weight decay | epochs | edit strength | text guidance | edit strength | text guidance |
| iWildCam | 0.0001 | 1e-4 | 100 | 0.4 | 5.0 | 1.3 | 7.5 |
| CUB | 0.001 | 1e-4 | 200 | 0.6 | 7.5 | - | - |
| Waterbirds | 0.001 | 1e-4 | 100 | 0.3 | 5.0 | 1.2 | 5.0 |

Table 6: Hyperparameters

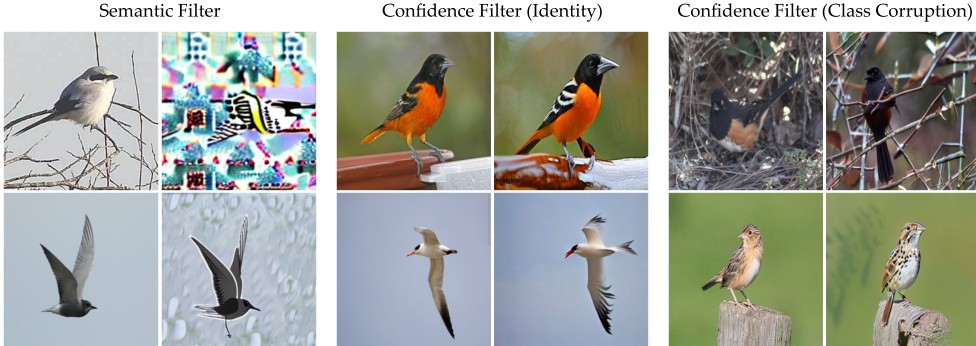

Figure 10: Random Sampled Edits filtered from Cub.

# 10    Additional Visualizations

## 10.1    Filtered Images

We present qualitative evidence confirming that our semantic and confidence filters effectively remove failed edits. Figure 10 showcases edits filtered from Cub with the semantic filter (left), the

confidence filter with high confidence and correct prediction (center), and the confidence filter with high confidence but incorrect prediction (right). As detailed in Section 3.3, the semantic filter removes total failures or irregular looking images, while the confidence-based filter eliminates edits that are excessively similar to the original image or that corrupt task-relevant information. Edits were created using Img2Img with the prompt "a photo of a { } bird perched on a branch."

## 10.2 Different Strengths and Text Guidance

The performance of ALIA in Section 5 is significantly influenced by the hyperparameters related to the image editing techniques, namely, the edit strength and text guidance.

The edit strength hyperparameter in the stable diffusion model controls the extent of modifications to the original image, with higher values yielding more noticeable and significant changes. We edited iWildCam images using the prompt "a cameratrap photo of a   near a large body of water in the middle of a field," employing Img2Img and InstructPix2Pix with edit strength ranging from $0.1 \rightarrow 0.9$ for Img2Img and $1.1 \rightarrow 1.9$ for InstructPix2Pix. Both Img2Img and InstructPix2Pix had text guidance set to 7.5. As illustrated in Figure 11, increasing the edit strength leads to greater departure from the original image, albeit at the risk of class information corruption.

The text guidance hyperparameter in the stable diffusion model determines how closely the generated output aligns with the textual description. Higher values produce outputs that more explicitly conform to the provided text. We edited iWildCam images using the prompt "a cameratrap photo of a { } near a large body of water in the middle of a field," using Img2Img and InstructPix2Pix with text guidance values of 5.0, 7.5, and 9.0. The edit strength was set to 0.4 for Img2Img and 1.3 for InstructPix2Pix. As Figure 12 demonstrates, higher text guidance results in images with more evident water, as specified in the prompt. However, similar to the edit strength hyperparameter, higher text guidance may also lead to greater class information corruption.

In practice, we found that selecting strength and text guidance values that yield more pronounced edits ultimately resulted in higher performance. This is because such edits significantly alter the context, while the filtering step mitigates the effects of detrimental edits. For example, running the same experiment as described in Section 5.1 with strength = 0.3 and text guidance = 5.0 resulted in an accuracy of 76.11%. This marks an approximate 8 point drop compared to strength = 0.6 and text guidance = 7.5 (84.87%).

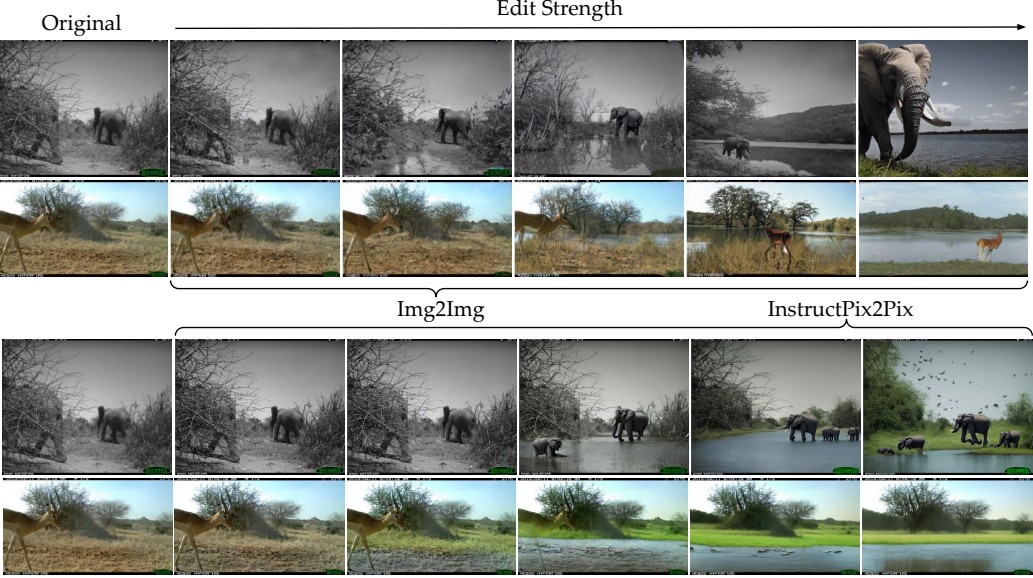

Figure 11: Different Edit Strengths.

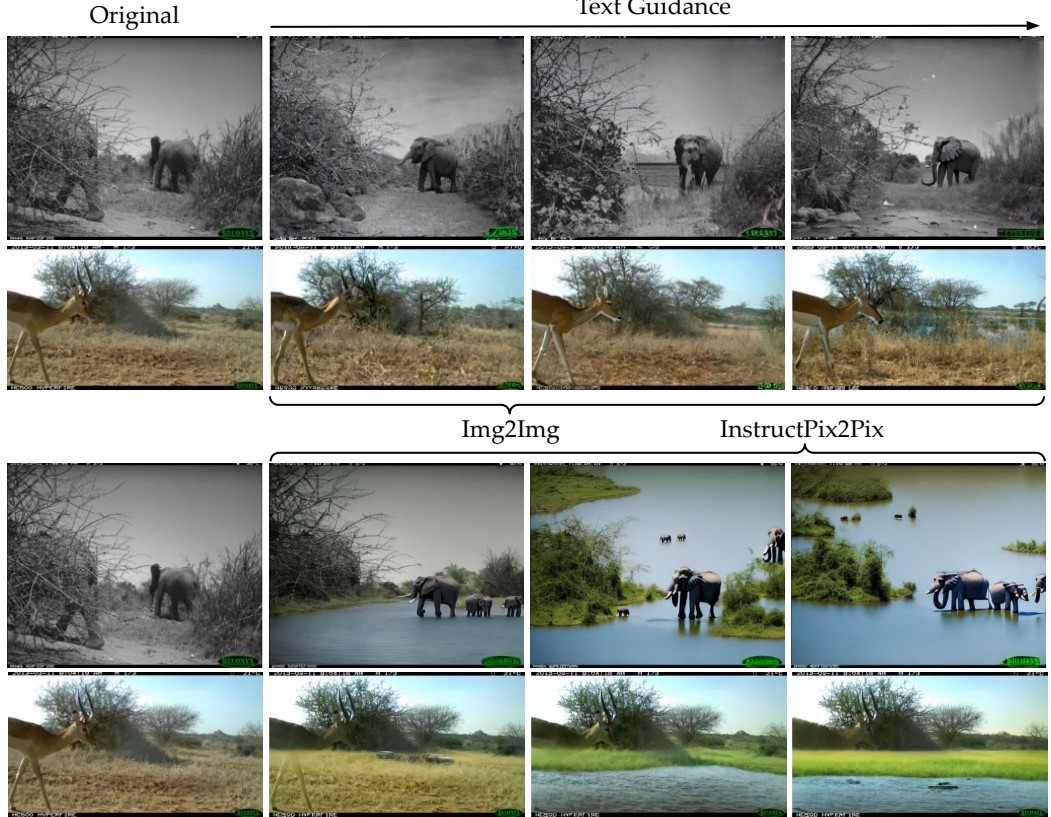

Figure 12: Different Text Guidance.

## 10.3 Generated Images

Figure 13 and Figure 14 show examples from each prompt used to create the ALIA augmented dataset (Section 5), before the filtering step. We also include samples of the poor quality edits produced by Img2Img on Waterbirds in Figure 15.

Img2Img

InstructPix2Pix

Txt2Img

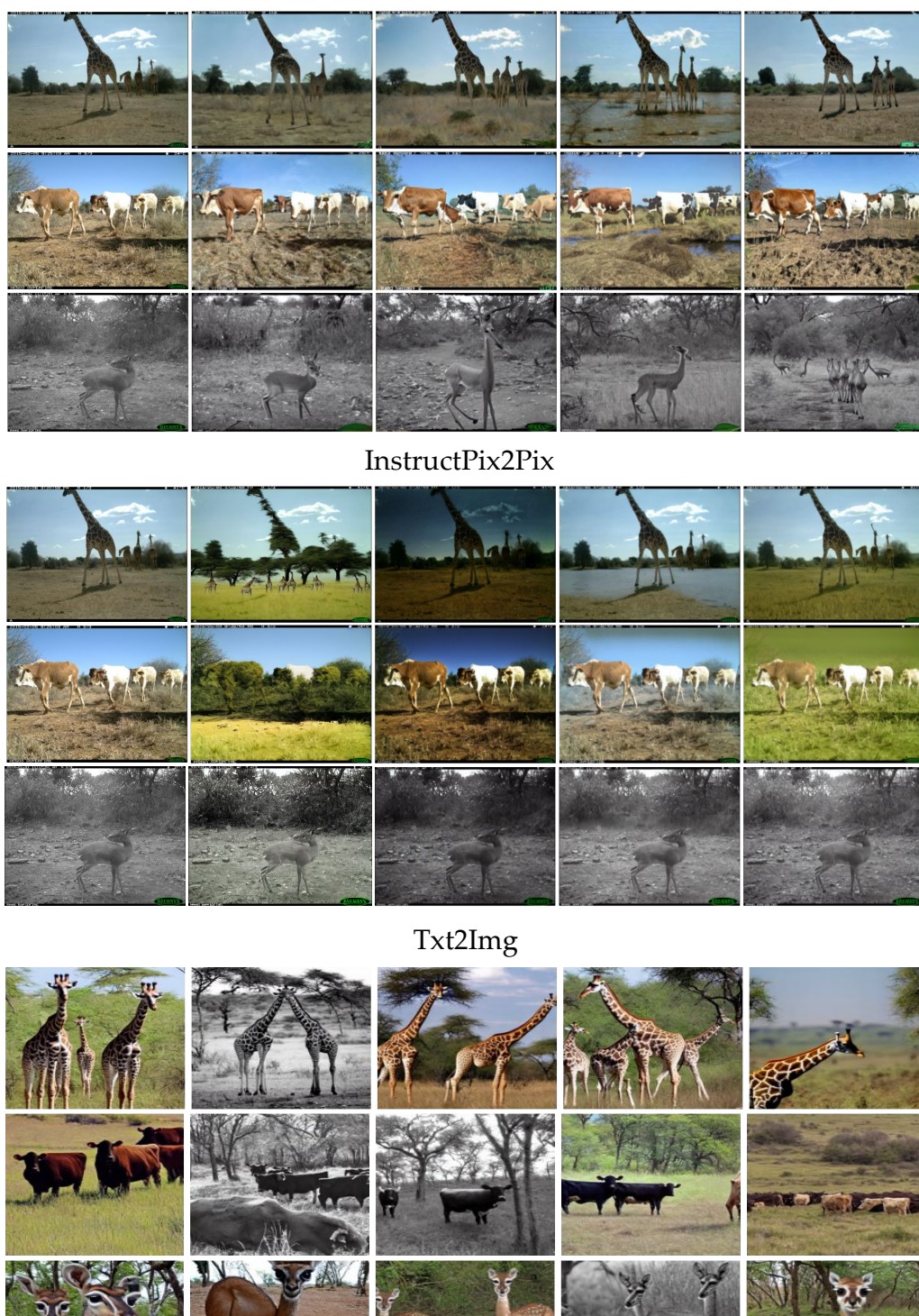

Figure 13: Randomly sampled edits of iWildCam, unfiltered. For Img2Img, the first column corresponds to the original image, and the further columns correspond to the following prompts (from left to right): *"a camera trap photo of a { }..."*: (1) *"in a grassy field with trees and bushes."*, (2) *"in a forest in the dark."*, (3) *"near a large body of water in the middle of a field."*, (4) *"walking on a dirt trail with twigs and branches."*. Txt2Img samples are generated with the prompt *"a camera trap photo of a { } in the wild."*.

Img2Img

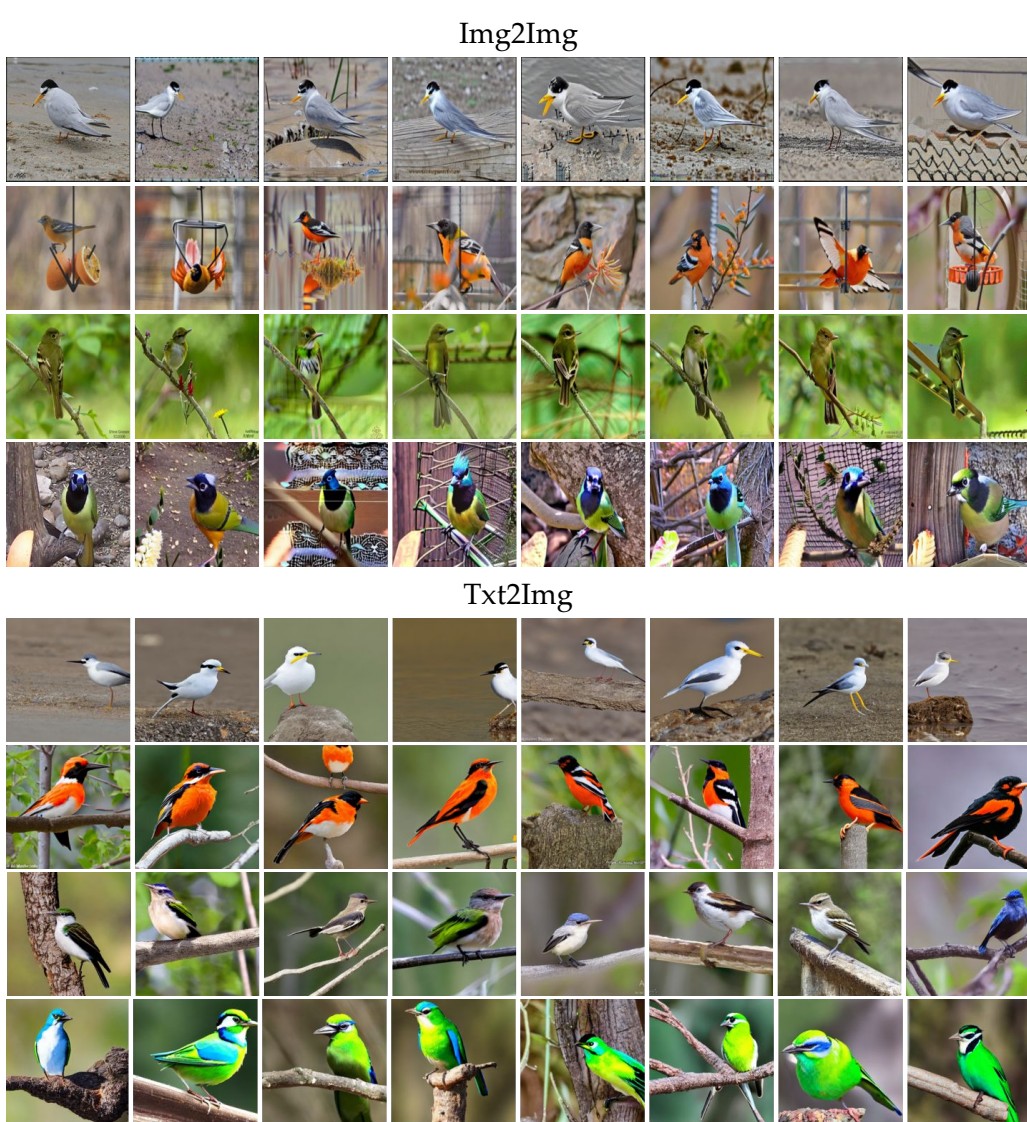

Txt2Img

Figure 14: Randomly sampled edits of Cub, unfiltered. For Img2Img, the first column corresponds to the original image, and the further columns correspond to the following prompts (from left to right): *"a photo of a { } bird..."*: (1) *"interacting with flowers."*, (2) *"standing by the waters edge."*, (3) *"perched on a fence."*, (4) *"standing on a rock."*, (5) *"perched on a branch."*, (6) *"flying near a tree, sky as the backdrop."*, (7) *"perched on a birdfeeder."*. Txt2Img samples are generated with the prompt *"an iNaturalist photo of a { } bird in the wild."*.

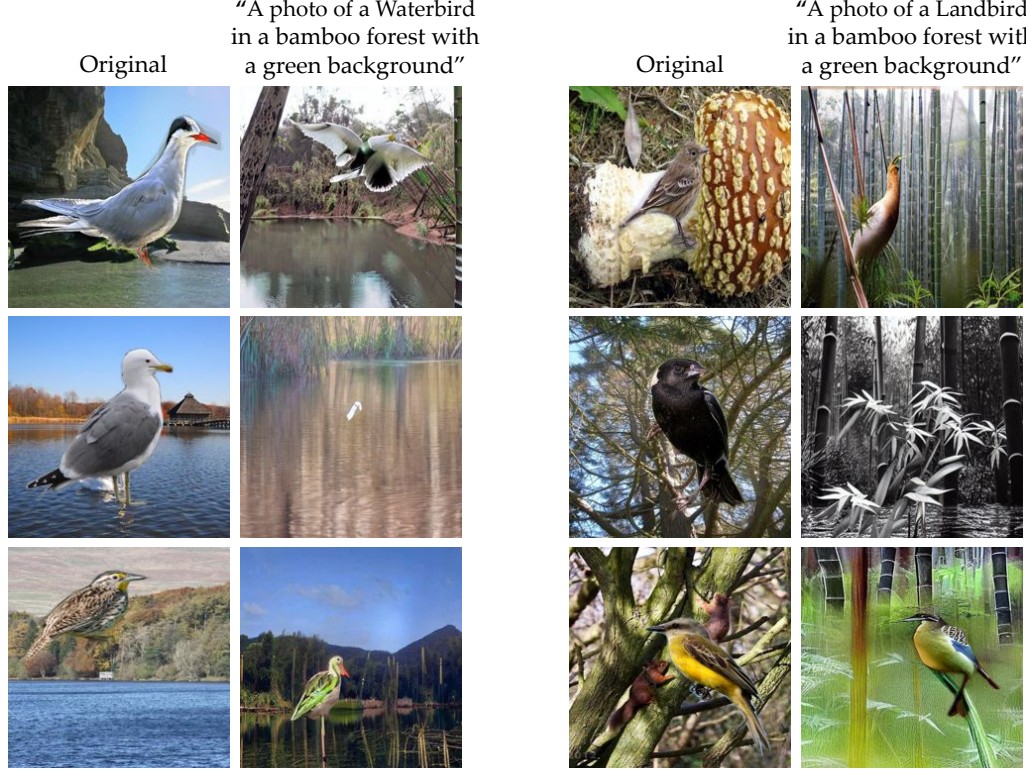

Figure 15: Randomly sampled edits of Waterbirds using Img2Img. Notice that in all cases the bird if heavily corrupted or removed.