# OpenReview forum: "Diversify Your Vision Datasets with Automatic Diffusion-based Augmentation"
_NeurIPS.cc/2023/Conference — NeurIPS 2023 poster_

### Official Review · Reviewer_Aa4b · 2023-06-27

**Soundness:** 3 good
**Presentation:** 3 good
**Contribution:** 3 good
**Rating:** 7
**Confidence:** 4

**Summary:**

The paper proposes a generative data augmentation method, ALIA, which utilizes large image captioning and language models to summarize the domain description and employs language-guided image editing methods to create augmented training data. Experiment results show that ALIA outperforms recent data augmentation methods on fine-grained classification tasks.

**Strengths:**

- S1. The paper is well-written and easy to follow.
- S2. The idea of using prompts to extract domain information and class-agnostic descriptions is novel.
- S3. The empirical result of the iWildCam experiment is promising. It is impressive that the task model trained on additional ALIA-generated images outperforms that uses real samples.


**Weaknesses:**

- W1. Limited technical contribution. Although the idea of using prompts to extract domain information and class-agnostic descriptions is novel, most building parts of ALIA are off-the-shelf methods, for example, BLIP, GPT-4, Img2Img and Instruct Pix2Pix.

- W2. The proposed method relies on pre-trained models heavily. As mentioned in the limitation section, ALIA is likely not as effective for unseen test domains. If ALIA performs poorly on a target dataset, no feedback can be sent to fine-tune ALIA on the dataset.

- W3. ALIA is only tested on three datasets, where the dataset classes are mostly similar (see Q1).


**Questions:**

- Q1. While being a generic data augmentation method, it seems that each tested dataset is comprised of similar objects, e.g., CUB contains birds, FGVC-Aircrafts contains aircraft and iWildCam contains animals. Does ALIA assume the dataset classes to be similar? On a more diverse dataset like ImageNet, can the large language model summarize good domain descriptions and create valid edits?

- Q2. In line 118. The authors constrain the number of domain descriptions to be less than 10. Can the authors explain more about the choice of the number of domain descriptions? How does it affect the quality of the augmented samples?

- Q3. In lines 191-192. The authors mention generating twice as much data as the original data to ensure “enough” data is available after filtering. How do we know whether the amount of filtered data is “enough” or not? Should we generate more data if the original dataset size is smaller?


**Limitations:**

I find no negative societal impact in this work.

---

> ### Author Rebuttal · Authors · 2023-08-09
>
> We appreciate your feedback and agree that our method is dependent on the ability of the pretrained models we use. We address your other questions and concerns below.
>
> **Datasets.** Our datasets consist of similar objects because we focused on the difficult setting of fine-grained classification. This choice was to see if diffusion could maintain the fine grained features of the class while changing the spurious features. That being said, ALIA does not need the classes to be similar, and could be applied to improve performance on datasets like ImageNet. Furthermore, we agree that more datasets are better and thus we added in an additional contextual bias dataset Waterbirds, which shows that ALIA outperforms everything but the real data baseline.
>
> **Choice of number of domain descriptions.** Our choice of less than 10 domain descriptions was due to compute constraints, since we generate two edits for each image in the dataset for each prompt. While the number of domain descriptions doesn't affect the quality of the augmented samples, having a really small number of prompts could potentially affect the diversity of the resulting augmented set.
>
> **Amount of data to generate.** The choice to generate twice as much data as the dataset (2 edits per image) is largely due to the real data baseline. Because we set aside real data to compare to (+Real), we needed all generated data techniques to add in the same amount of data per class as this real data baseline. In practice one could edit a small percentage of their dataset and see what percentage of images were filtered out, then use that to determine roughly how many images to edit in order for the filtered dataset to be of the desired size. For example, if one wanted to add in 10 samples per class and profiling on a small generated dataset shows that around 50% of the images get filtered out, at least 20 images should be edited per class.

---

> > ### Comment · Reviewer_Aa4b · 2023-08-18
> >
> > Thanks for the reply. The authors addressed most of my concerns by providing additional results on the Waterbirds dataset and insights on choosing different numbers of domain descriptions and the amount of data to generate. In summary, the proposed method demonstrates a good utilization of LLM and generative models for data augmentation. I decided to increase my score from 6 to 7.
> >
> > In the final version, I recommend the authors include additional experiments to validate that the number of domain descriptions doesn't affect the effectiveness of the augmented samples.

---

> > > ### Author Response · Authors · 2023-08-18
> > > **Thank you**
> > >
> > > We are glad we have addressed most of your concerns. We agree that an ablation on the number of domain descriptions would provide useful insight into the method and will strive to include those results by the camera ready.

---

### Official Review · Reviewer_JLbC · 2023-07-04

**Soundness:** 3 good
**Presentation:** 3 good
**Contribution:** 3 good
**Rating:** 7
**Confidence:** 4

**Summary:**

The paper introduces ALIA to generate domain descriptions of the dataset from captions of each image. These captions are then used to generate more data using Stable Diffusion. There is also a filtering process to remove corrupted images or those with minimal edits. ALIA improves performance over the baselines on several datasets.

**Strengths:**

- The paper was well written and easy to follow.
- Fine-grained classification is an interesting and difficult problem. It is also nice to see a different setting compared to existing papers trying to improve robustness on IN-Sketch/R etc. Furthermore, the method performs well compared to the baselines.
- I like the discussion on filtering failed edits. It is important but usually not given alot of attention.

**Weaknesses:**

- Data and computational efficiency
    - How many new images were generated for the baselines and ALIA? It would be useful to have a plot of how the performance of the methods changes with generated data and compute time. Augmentation methods like CutMix, RandAug have the advantage of being relatively efficient and more scalable than generation methods. This may matter if the difference in accuracy is not as large e.g. between RandAug and ALIA in Fig 5.
- Background domain from test images
    - ALIA seems to be informed of the test domain, at least for iWildCam, while the other baselines are not. Domain adaptation methods that makes use of unlabelled target images, or test time adaptation methods may be more relevant baselines.

**Questions:**

Other than the questions above,
- There are existing datasets with contextual bias e.g. Waterbirds, CelebA. I was wondering why the authors chose to create one with the Aircraft benchmark.
- Is it interesting that img2img does not work well for the Aircraft dataset, do the authors have any intuition why?

**Limitations:**

Yes

---

> ### Author Rebuttal · Authors · 2023-08-09
>
> Thank you for your insightful review. We hope that we have adequately addressed your concerns below:
>
> **Data and Computational Efficiency.** We generate 2 edits per image per prompt in the training set for ALIA. Depending on the output of the LLM, the number of prompts vary from 4-10. Using the HuggingFace implementation of Img2Img editing on a GeForce RTX 2080 Ti, we can produce 2 edits of a given image in approximately 2 seconds, resulting in a generation time of around 15 minutes for the smallest dataset (Airbus VS Boeing) and around 7 hours for the largest dataset (iWildCam Subset). We will include the time it took to generate our datasets in the appendix. While our method does require a lengthy generation time if the dataset is large, our primary focus in this paper was to test the quality of our augmentations rather than the efficiency.
>
> **Background Domain from Test Images.** For iWildCam we do use unlabeled test images to generate the domain prompts for editing, but crucially we only use images from the background class, so we do not obtain any data of the animals in the test domain. While there do exist domain adaptation baselines which can make use of this test data, our paper is focused on data augmentation methods specifically.
>
> **Choice of Contextual Bias Benchmark.** We chose the Aircraft dataset because it is a challenging fine-grained dataset, with contextual bias that wasn't artificially created like Waterbirds. However, we do have results for the Waterbirds dataset, which are included in the global response and pdf. As you can see, ALIA outperforms all baselines aside from adding real data in the case of 100% contextual bias.
>
> **Img2Img Failure on Aircraft dataset.** Our hypothesis as to why img2img does not work well for the Aircraft dataset is because of the lack of patterns in the image. Many of the planes are displayed against a uniform blue background and are mostly white themselves. We saw a similar effect when trying to edit simple cartoons, or plane pictures taken randomly from Google. That being said, we predict that this problem will go away with better diffusion models or models trained on more images similar to the plane images. We will add this explanation to the appendix.

---

> > ### Comment · Reviewer_JLbC · 2023-08-17
> > **Thanks for the clarifications!**
> >
> > Thanks for the additional results on waterbirds, runtime and clarifications. It is nice to see a paper focusing on non ImageNet tasks. I also appreciate the insights on how edits can affect datasets differently and the discussion on filtering failed edits. I have read the other reviews and responses too and have decided to increase my score to 7.

---

> > > ### Author Response · Authors · 2023-08-18
> > > **Thank you**
> > >
> > > We are delighted that our rebuttal addressed all your questions/concerns and appreciate your raised score!

---

### Official Review · Reviewer_p3WR · 2023-07-06

**Soundness:** 3 good
**Presentation:** 4 excellent
**Contribution:** 4 excellent
**Rating:** 7
**Confidence:** 4

**Summary:**

The paper proposes a method to augment an existing vision dataset with samples that are likely reflective of task-relevant variations that are potentially missing in the same. To do this, the authors propose a method, ALIA (Automated Language-guided Image Augmentation), which utilizes off-the-shelf image-generation and language model pipelines to first generate diverse domain descriptions using a captioning + LLM pipeline which can then be used to edit an existing image to generate diverse variations of the same. Specifically, ALIA consists of three steps – (1) prompting an LLM to generate domain descriptions (likely capturing visual variations) based on a collection of generated captions associated with the original dataset, (2) using an image editing pipeline (Img2Img / Instruct Pix2Pix) to generate edits and (3) automated filtration steps to ensure the generated images and visually consistent and can reliably augment the existing dataset. From experiments conducted across three benchmarks – domain generalization (iWildCam), fine-grained classification (CUB) and classification in the presence of contextual bias (custom split of FGVC-Aircraft) – the authors show that ALIA guided expansion of the data distribution is most-effective in improving performance over a vanilla baseline and other prior augmentation strategies considered. Additional ablations outline the extent to which prompting, filtration and the editing mechanism impact performance.

**Strengths:**

The following points outline the strengths associated with the submission.

- The paper is generally well-written and easy to follow. The authors do a good job of motivating the base observations – (1) circumventing additional data curation by using generated images and (2) adopting conditional text-guided edits as a more structured way to generate samples to augment an existing dataset. The introductory section does a good job of outlining the necessity of individual components in ALIA – the necessity of invoking task-agnostic language descriptions to guide edits, avoiding fine-tuning of individual large-scale models and including an automated filtration step.
- In my opinion, compared to prior work (as noted by the authors as well), the novelty of the proposed approach lies in automating the pipeline to generate augmented samples with minimal interventions. While it remains to be seen the extent to which “avoiding fine-tuning” will translate to more complex settings (images with multiple objects)  and tasks (structured prediction), ALIA seems like a novel step in this direction of expanding data distributions by ensuring broad coverage of variations likely to be seen in test-time settings (modulo the obvious caveats acknowledged by the authors in Section 6).
- The proposed method seems to work and leads to improvements over a vanilla baseline, other diffusion-guided (conditional / unconditional) editing schemes and prior augmentation strategies, and is often competitive with an oracle setting where one has access to data from the test distribution. With the exception of the points raised under weaknesses, the significance of ALIA lies in the fact that it is a straightforward and timely combination of existing techniques in language modeling and image generation / editing / personalization, and works fairly well off-the-shelf across multiple settings. Additionally, I particularly like the filtration process – designed by first identifying potential failure modes and subsequent methods to counter those.
- The ablations (coupled with underlying hypotheses wherever applicable) are useful and provide actionable insights about the extent to which components in ALIA are sensitive to the data and the model at hand – for instance, while InstructPix2Pix is more useful compared to Img2Img, in the contextual bias settings, it leads to artifacts for iWildCam edits. These observations are likely going to be useful for future work attempting to build on top of ALIA.


**Weaknesses:**

The following points outline the weaknesses associated with the submission. Most of these points are either associated with the significance and completeness towards the intended goal.

- Given that the description of a “domain” is left somewhat open-ended in the current version (L83-85), the paper would benefit from including the discussion (supported perhaps by quantitative results) surrounding – (1) what kind of semantic / stylistic variations are missing in the base-dataset and (2) whether ALIA explicitly counters those scarcities by introducing relevant edits. While the examples provided for the contextual bias settings in Figure 7 help highlight how over-represented settings are pruned (and under-represented ones are highlighted), a discussion surrounding all different settings would significantly strengthen the submission.
- Following up on the previous point, in addition to above, including an analysis informing the extent to which one needs to augment an existing dataset would be useful as well – is it necessary to consider an expansion of 20-100% in most settings? It may not matter as much for small-scale settings, but for someone intending to build on top of ALIA, it may be useful to know if one runs into a “diminishing return” scenario for different datasets at any point – i.e., if gains obtained by adding more diverse samples become increasingly marginal beyond a point.
- L215-216 states that domain descriptions for the background images from the test-set were used to generate augmented samples in iWildCam. Coincidentally, iWildCam is the only setting where +ALIA > +Real. Since assuming access to the specific “kind” of target domain variations during training is not entirely fair (one does not know the test-time variation apriori), the paper would benefit from relaxing claims surrounding “beating real data” for this specific setting.
- For the contextual bias experiments, it might be beneficial to consider settings where spurious correlations have been studied heavily (for instance, the NICO [A] benchmark). Not only would that involve a more diverse “base” dataset, but also help compare with more sophisticated algorithms (a subset of which are designed to explicitly counter spurious correlations). This is motivated from the fact that supplementing with “targeted” data may not always be the best solution in contextual bias settings.


[A] – Towards Non-I.I.D. Image Classification: A Dataset and Baselines


**Questions:**

The points outlined under strengths and weaknesses influence my rating. Regarding weaknesses, my suggestions are intended more towards improving completeness and significance of the results presented in the current submission. Among these, I think (1), (3) and (4) are crucial weaknesses which are perhaps central to the claims of the paper. Addressing these would definitely help me in improving my rating of the paper.

**Limitations:**

The only potential negative societal impact that I foresee is that models trained on ALIA augmented datasets can “unintentionally” inherit biases present in the base models (LLM, Diffusion) in the ALIA pipeline. While this may not matter as much for the datasets being considered for experiments in the submission, it is perhaps crucial for other work attempting to build on top of ALIA for sensitive situations. Adding a discussion (with obvious disclaimers) would improve the current submission.

---

> ### Author Rebuttal · Authors · 2023-08-10
>
> Thank you for your thoughtful feedback and suggestions, we hope to have addressed each of your concerns with the following:
>
> **More clarity around domain shift.** We agree that more clarity around what each domain shift is, in addition to quantitative proof that ALIA improves accuracy under this domain shift, is crucial to showing the efficacy of our method. Below we describe the shifts in each dataset (if available) as well as proof that ALIA improves performance in each of these cases. Note that Cub2011 has no explicitly defined domain shift.
> * *iWildCam:* This dataset is constructed such that the locations of the camera trap differ from train to test; specifically, the locations containing (1) a lake in the distance or (2) a dirt trail with trees are two locations not present in the training set. A sample of images from these locations are included in the global rebuttal PDF. We see that our prompt generation technique does produce domain descriptions that describe the test domain, such as “a camera trap photo of a {} near a large body of water in the middle of a field.”, “a camera trap photo of an {} in a forest in the dark.”, and “a camera trap photo of a {} walking on a dirt trail with twigs and branches.” We further see in Figure 4 of the paper that when evaluating on these two locations, we get high performance compared to other approaches.
> * *Airbus VS Boeing:* The domain shift in this dataset is the existence of Airbus planes on grass and Boeing planes on road in the test set. More explicitly, examples considered in-domain are Airbus on road, Boeing on grass, Airbus in the sky, and Boeing in the sky, which appear in both the training and test set. An exact breakdown of the number of samples in each group are in the Appendix. As shown in the global response and PDF, our method is able to improve on the in-domain performance of all augmentation methods, while also beating the baseline and traditional data augmentation methods on the out-of-domain performance.
> * *Waterbirds:* as described in the global rebuttal, the domain shift here is the presence of Landbirds on Water and Waterbirds on Land in the test set. As shown in the global response and PDF, our method is able to roughly match the in-domain accuracy of the other methods while drastically improving the accuracy of all other augmentation methods.
>
> **Amount of Generated Data VS Accuracy.** Although we did not have time to investigate this for all datasets, we were able to experiment on how accuracy changes with the amount of generated data added on Cub2011. As shown in the results (contained in the global rebuttal PDF), ALIA is able to achieve accuracy gains up to 1000 images (20% of the original training set), at which point accuracy starts to decline. In contrast, images generated from text alone see a decrease in accuracy almost immediately, reaching below the accuracy of the original training set at 2000 images. We will work to get similar plots for the other datasets by the camera ready.
>
> **Claims of beating real data.** We agree that we should provide more context to the claim of ALIA beating real data, especially considering that we do have knowledge of the test domain. We will update our manuscript accordingly.
>
> **Spurious correlation datasets.** While we agree that NICO is a more diverse benchmark that simulates real world domain shifts, we chose not to use it in our evaluation because the images seemed as though they could be easily mimicked with Stable Diffusion. In these cases, the best approach would likely use Txt2Img data, as it is more diverse than image editing.

---

> > ### Comment · Reviewer_p3WR · 2023-08-14
> > **Thanks for the response!**
> >
> > Thanks for responding to my concerns and providing additional experimental results. My primary concerns surrounding the shifts ALIA tackles and the impact of the amount of generated data were adequately addressed by the rebuttal.
> >
> > Most concerns from other reviews seem to have been addressed (with supporting discussion as well) adequately as well. I would encourage the authors to include (additional) discussions from the reviews in the revised version.

---

> > > ### Author Response · Authors · 2023-08-18
> > > **Thank you**
> > >
> > > We are glad that our rebuttal resolved the concerns you had. Thank you for the suggestion of analyzing the amount of generated data added; we think it has provided much more clarity on our method.

---

### Official Review · Reviewer_za7y · 2023-07-07

**Soundness:** 4 excellent
**Presentation:** 3 good
**Contribution:** 3 good
**Rating:** 7
**Confidence:** 4

**Summary:**

This paper explores improving robustness to variations in domains with the use of modern large vision and language models. They propose ALIA (Automated Language-guided Image Augmentation) which generates automated descriptors of data domains and augments training data with language-guided image editing. They use a model trained on the original dataset as a quality filter to ensure class-relevant information is maintained. They show this approach leads to nice gains on fine-grained datasets or data with significant clutter, for both classification and detection.

After reading the authors rebuttal and discussing with them, I will maintain my score of 7.

**Strengths:**

It’s clever to combine recent large models in this way, using image captioning to generate appropriate background or context caption templates across based on what is seen in the dataset, and allowing language-guided editing to provide diversity in a more realistic manner than randaug or cutmix based on those templates. In particular, the use of semantic features and visual features (via classifier confidence) to filter failures both where the image is only minimally changed or where the image is changed in a way that corrupts class information is quite nice. The experiments, across several datasets and dimensions of challenges including domain shift and contextual bias, were nicely done and visualizations were clear and informative.

**Weaknesses:**

Since the domain descriptions are build with the existing dataset, it seems that this method may not work as well for cases where the dataset in question is not representative of all potential domains seen in practice, or where the dataset has significant bias or gaps. Additionally, since the diverse augmented data is conditioned on training examples and the category in question often is minimally changed, there is still limited diversity for rare species with few examples during training, particularly rare pose diversity. It would be nice to explore this more deeply on a large-scale fine grained dataset with significant imbalance, such as iNaturalist.

The biggest weakness of this paper is lack of clarity and transparency about some of the data choices made. In particular, it wasn’t clear what protocols were used when selecting subsets of iWildCam for train and test.

**Questions:**

Could performance be further improved by including domain descriptors for test data (assuming access to test data at inference time, but not test labels) for cases with domain shift between train and test?

In some cases, context is highly important and valuable species identification information for human experts, and certain species would not ever been seen, eg, perched on a birdfeeder, out at night, in the water. Have you considered learning class-appropriate subsets of language templates per-species, to reduce potentially confusing and unrealistic generated context?

I noticed that you considered only a subset of iWildCam classes and camera locations, what was the reasoning behind the choice of subsampling? Did you artificially balance the dataset at all (ie removing rare categories or capping common ones) or did you keep a realistic shift in subpopulation distribution across different domains? How were the real test-domain images sampled for comparison, uniformly across the test camera sites and categories of interest or?, Did you make sure that the test data was identical when comparing (ie remove that added test data from evaluation for all models)? It would be good to be very explicit about all of these choices whenever you use a non-standard split of a dataset

Does performance ranking across baselines change if we consider different metrics? For example, if we consider top-1 accuracy or break down performance per class on iWildCam instead of looking at the macro f1? Does this method improve more for rare classes or common ones?

**Limitations:**

The limitations section is well done, and points out several clear potential failure modes of this method. More detailed analysis of failure cases would be appreciated. For which classes does this method help, and for which does it hurt? Are there any consistent patterns in data that the model finds difficult to predict accurately, even after augmentation?  It would help the reader build additional intuition for where gaps remain and in which cases this method is ready for deployment.

---

> ### Author Rebuttal · Authors · 2023-08-09
>
> We appreciate your appreciation of our work! We agree that ALIA is not suitable for cases were there are unseen domains which are significantly different from the domains seen in training or cases where the class relevant features need to be significantly augmented (e.g. pose change). As we will detail below, our subset of iWildCam does include class imbalance, but since the animals are mostly well represented by Stable Diffusion, we are unsure of how well ALIA can handle class imbalance for rare species. Due to time constraints we were not able to test ALIA on iNaturalist, but we will strive to have results for the camera ready.
>
> **Including Domain Descriptors for Test Data.** Yes! For our iWildCam experiment, we did exactly this: took the unlabeled background images for the new domains and used those to generate the domain descriptions. This is an especially useful tool in cases of a large domain shift like sim to real transfer, and the user can also explore using their own constructed prompts if they anticipate a particular test domain but have no data for it.
>
> **Class-Specific Domain Descriptions.** This is another great idea that we have been exploring. For instance, one could generate domain descriptions per class and use these descriptions to generate diversity within a class, or inversely could use these to uncover potential biases in the dataset and direct the augmentation to a subset of classes. There are a lot of interesting avenues around selecting which images to augment with which prompts, and we see this as exciting future work.
>
> **iWildCam Subset Construction.**
> We apologize for not being clearer on how and why this dataset was constructed. We chose to construct a subset of iWildCam because (1) the experimentation and generation time for the entre wilds dataset was prohibitively expensive and (2) some classes didn’t have enough examples for us to split into a train, val, test, and extra set. While we did not artificially balance the classes, we did want classes which would have at least 40 examples in each split.
>
> In an effort to keep this constrained setting as close to the original iWildCam dataset as possible, we constructed the train/val/test/extra splits such that each split contains non-overlapping locations and we did not subsample within locations to balance the class support.
>
> Since iWildCam has a train, id_test, and test split already, we sampled the locations with at least two classes to put into our subset, ensuring that the train subset was sampled from the train set, the val subset was sampled from the id_test set. We split the sampled locations from the iWildCam test set into two disjoint groups, which formed the test set and extra set for our subset. These new splits were set before experimentation, so all methods were trained, validated, and evaluated on the same data. We have included a breakdown of the class counts of each split below. We will also include this description in our final manuscript. Please let us know if you have any further questions surrounding the construction of this dataset.
>
> | Split | Background | Cattle | Elephant | Impala | Zebra | Giraffe | Dik-Dik |
> | :---  |:----: |:---: | :----: | :---: | :---: | :----: | :---: |
> | Train | 2210 | 801 | 366 | 981 | 720 | 460 | 514 |
> | Val   | 2006 | 206 | 53 | 140 | 119 | 58  | 244 |
> | Test | 127 | 1416 | 2003 | 4553 | 144 | 47 | 213 |
> | Extra | 402 | 506 | 91 | 625 | 85 | 47 | 468 |
>
> **iWildCam Metrics.** Below we include the accuracy for our iWildCam experiments, which again shows that our method outperforms all baselines as well as real data. Since we do not add in the same number of images per class in our experiment, it would be hard to draw useful conclusions on which classes our method does well for, but our intuition is that the edited data for classes which stable diffusion cannot recreate well will likely be of lower quality even after filtering, and thus may result in worse performance. We will strive to provide further quantitive analysis on the failure modes of ALIA by the camera ready.
>
> | Dataset  | Accuracy          |
> | :---     |   :---:           |
> | Baseline | 67.51(6.15)       |
> | +CutMix  | 77.56(4.77)       |
> | +RandAug | 72.97(3.87)       |
> | +Txt2Img | 75.59(3.36)       |
> | +ALIA    | **84.87(1.92)**    |
> | +Real    | 74.10(3.37)       |

---

> > ### Comment · Reviewer_za7y · 2023-08-17
> > **Thank you for the clarifications**
> >
> > Thank you to the authors for the clarifications on data subset construction. This line of research on improving specialized OOD performance with augmentation from generated images is distinct from the more common in-distribution challenge presented by datasets like ImageNet. This work demonstrates a nice step forward in exploring what works well for OOD diversification, despite restricting the realism and possibly usefulness of the test case by focusing on common categories that are well represented in Stable Diffusion (I would still be very interested in seeing iNaturalist results in the camera ready). Because of this, I will keep my score of 7 for this work.

---

> > > ### Author Response · Authors · 2023-08-18
> > > **Thank you**
> > >
> > > We are glad that our rebuttal resolved your questions. We are currently working on implementing ALIA for iNaturalist and will strive to have the results for the camera ready!

---

### Official Review · Reviewer_u22M · 2023-07-09

**Soundness:** 4 excellent
**Presentation:** 3 good
**Contribution:** 2 fair
**Rating:** 3
**Confidence:** 4

**Summary:**

This paper presents a data augmentation methods. It first leverages the advanced image captioning and large language models to generate  prompts. Next, it uses diffusion-based models to generate more images. In addition, it trained a model to filter failure or similar cases during image generation. Extensive experiments over 3 specialised tasks demonstrate the effectiveness of the proposed method.

**Strengths:**

1. This paper is general clear and easy to follow
2. The proposed method is simple and technically sounds
3. The proposed method effective improves the classification performances over the reported three tasks

**Weaknesses:**

This paper studied the generated images through diffusion models on "specialised" classification tasks. However, generated images through diffusion models have been proved to be effective on improving "general" classification tasks [8, 35]. Therefore, the novelty and contribution of the proposed method seems to be incremental.

**Questions:**

1. It have been studied in [8, 35] that images generated from diffusion models could help to improve the image classification performances. The author claims that "we use diffusion models to do image editing with text rather than generating images from text alone, resulting in augmentations that closely resemble the training data without finetuning." However, I'm thinking stable diffusion models (text2image models) could generate more diverse data as compared to text editing diffusion models (e.g. InstructPix2Pix). Then why text editing methods are better? Experimental results to compare the augmented data through text2image and text editing methods are expected.
2. This paper aims to improve the performances on specialised tasks that have very few training data. However, some of these tasks may contain out-of-domain data that may be difficult to be generated through diffusion models. Discussion regarding to these cases are expected.
3. It is mentioned "We fine-tune a ResNet50 for the CUB [40] and iWildCam [15] datasets and a ResNet18 for the Planes [21] dataset". Why use different networks for different datasets?
4. In Appendix, the generated images through img2img of Airbus and Boeing look very strange. Discussion regarding to these results are expected.

**Limitations:**

Yes

---

> ### Author Rebuttal · Authors · 2023-08-09
>
> We appreciate your review and you bring up important questions on how our work differs from the several works that explore using diffusion generated data to improve "general" classification tasks. We aimed to address this in the introduction of our paper and we hope to provide more clarity below.
>
> **Motivation for Image Editing/OOD data.** We agree that text2image models produce more diverse images than image editing methods like InstructPix2Pix, and thus in a setting where the training/test images are well represented in the training set of Stable Diffusion, using images generated from text alone is likely to result in bigger gains than edited data.
>
> While this is an exciting avenue to explore, we are interested in how well these text conditioned generative models can do when the task is out-of-domain; that is, when text to image models cannot produce images similar to those in the training set. In these settings, it is unclear if/how one can utilize these powerful models to improve accuracy. The key insight of our work is that grounding the augmentation in the training data itself is an effective method of utilizing the high domain-level knowledge of Stable Diffusion (e.g. background, weather, lighting, etc) to vary existing images, thus maintaining the class-relevant features. As shown qualitatively, images generated from text alone look dissimilar to real training images while image edits result in images that are less diverse but more similar to the types of images seen in this specialized task. On datasets of various sizes, we show quantitatively that this results in an augmentation technique that beats both traditional augmentation strategies as well as using Txt2Img for augmentation.
>
> **Why use ResNet18 for Airbus VS Boeing.**
> We chose ResNet18 because the dataset is rather small and it made the experiments much less compute intensive. We will update the manuscript with the ResNet50 results for the camera ready.
>
> **The generated images for Airbus and Boeing look very strange.**
> Our hypothesis as to why img2img does not work well for the Aircraft dataset is because of the lack of patterns in the image. Many of the planes are displayed against a uniform blue background and are mostly white themselves. We saw a similar effect when trying to edit simple cartoons, or plane pictures taken randomly from Google. That being said, we predict that this problem will go away with better diffusion models or models trained on more images similar to the plane images. We will add this explanation to the appendix.

---

> > ### Comment · Reviewer_za7y · 2023-08-17
> > **general vs specialized**
> >
> > Reviewer u22M, I'm curious how you define the difference between "general" classification tasks and "specialized" classification tasks?

---

### Author Rebuttal · Authors · 2023-08-09

We thank all reviewers for their detailed reviews. We are delighted that they found the paper easy to follow and saw value in our discussions surrounding failed edits and different data augmentation techniques.

As requested, we have run a few more experiments to ensure a comprehensive evaluation. We have plotted the resulting data tables and included all plots in the attached PDF.


### Waterbirds [Contextual Bias]

Waterbirds[1] is a synthetically created dataset which creates contextual bias by taking species of landbirds and waterbirds from the CUB-200[2] dataset and pasting them onto forest and water backgrounds from the Places[3] dataset. For the training and validation sets, all landbirds appear on forest backgrounds and all waterbirds appear on water backgrounds, while the test set has an even representation of backgrounds and bird types. Further experimental details are contained in the PDF.

As shown below and in Figure 1 of the PDF, ALIA is able to roughly match the in-domain accuracy of the other methods while drastically improving the accuracy of all other augmentation methods. Note that we don't bold the +Real numbers in the table because it is considered an oracle baseline.

| Dataset | ID Accuracy | OOD Accuracy | Class Balanced Accuracy |
| :---        |   :----:   |   :---: | :----:   |
| Baseline      | 97.34(0.23) | 27.14(3.06) | 62.24(1.58) |
| +CutMix   | **98.01(0.33)** | 28.84(5.29) | 63.42(2.49)  |
| +RandAug | 97.61(0.51)  | 30.32(6.31) | 30.32(6.31) |
| +Txt2Img | 97.61(0.11) | 29.66(3.81) | 63.64(1.92) |
| +ALIA | 96.17(0.47) | **46.63(3.96)** | **71.40(1.84)** |
| +Real | 91.45(0.45) | 89.40(0.49) | 90.43(0.47) |

[1] Sagawa, et al. "Distributionally Robust Neural Networks for Group Shifts: On the Importance of Regularization for Worst-Case Generalization"

[2] Wah, et al. "The Caltech-UCSD Birds-200-2011 dataset"

[3] Zhou, et al. "Places: A 10 million image database for scene recognition"



### Number of generated images VS accuracy on Cub2011

While our main experiments restrict the number of generated images added to the dataset, one of the benefits of diffusion generated augmentation is that one can create infinite amounts of data. Figure 2 in the PDF depicts the accuracy on CUB as a function of the number of generated images added (200/400/1000/2000/4000), where the grey line is the baseline accuracy on the original 4994 images.

ALIA is able to achieve accuracy gains up to 1000 images (20% of the original training set), at which point accuracy starts to decline. In contrast, images generated from text alone see a decrease in accuracy almost immediately, reaching below the accuracy of the original training set at 2000 images. We suspect this is because much of the text to image data is from a different distribution than the regular training data, and thus adding small amounts can increase robustness while large amounts cause the model to overfit to this distribution. Since image to image edits use language to edit the training images directly, these images are less likely to be out of distribution as compared to the text to image data.

### Airbus VS Boeing ID/OOD accuracy breakdown

In order to show that ALIA does improve accuracy on the unseen domain, we break down the results from Figure 6 of the main paper into in-domain accuracy and out of domain accuracy. Note that we don't bold the +Real numbers in the table because it is considered an oracle baseline.

As shown below as well as in Figure 3 of the PDF, ALIA is able to improve on the in-domain performance of all augmentation methods while beating the baseline and traditional data augmentation methods on the out-of-domain performance.

| Method   | ID Accuracy        | OOD Accuracy       | Class Balanced Accuracy |
| :---     |   :---:            |   :---:            |   :---:                  |
| Baseline | 85.79(1.99)        | 31.68(0.60)        | 64.64(0.75)              |
| +CutMix  | 85.96(1.30)        | 32.11(1.50)        | 64.96(1.35)              |
| +RandAug | 87.23(1.07)        | 30.17(0.09)        | 64.87(0.65)              |
| +Txt2Img | 81.52(1.45)        | **41.62(0.21)**   | 65.85(1.12)              |
| +ALIA    | **88.11(0.84)**       | 36.20(0.36)        | **68.84(0.76)**              |
| +Real    | 87.07(1.16)        | 44.34(2.25)        | 71.81(1.29)              |

---

### Decision · Program_Chairs · 2023-09-21

**Decision:**

Accept (poster)

**Comment:**

The reviewers did not reach a full consensus. However, the main criticism of the rejecting reviewer is that it is already known diffusion models can generate images which help training. Indeed, this is shown as a baseline in the paper and the paper proposes editing training images instead. The AC finds this different and interesting enough to warrant publication.